# Genetic Diversity of *Bemisia tabaci* (Gennadius) (Hemiptera: Aleyrodidae) Colonizing Sweet Potato and Cassava in South Sudan

**DOI:** 10.3390/insects11010058

**Published:** 2020-01-17

**Authors:** Beatrice C. Misaka, Everlyne N. Wosula, Philip W. Marchelo-d’Ragga, Trine Hvoslef-Eide, James P. Legg

**Affiliations:** 1Department of Agricultural Science, School of Natural Resources and Environmental Sciences, University of Juba, P.O. Box 82, Juba, South Sudan; beatricelangwa@gmail.com (B.C.M.); drwani49@gmail.com (P.W.M.-d.); 2Department of Plant Sciences, Norwegian University of Life Sciences (NMBU), P.O. Box 5003, 1432 Ås, Norway; 3International Institute of Tropical Agriculture, P.O. Box 34441, Dar es Salaam, Tanzania; e.wosula@cgiar.org (E.N.W.); j.legg@cgiar.org (J.P.L.)

**Keywords:** *Bemisia tabaci*, genetic diversity, distribution, haplotype

## Abstract

*Bemisia tabaci* (Gennadius) is a polyphagous, highly destructive pest that is capable of vectoring viruses in most agricultural crops. Currently, information regarding the distribution and genetic diversity of *B. tabaci* in South Sudan is not available. The objectives of this study were to investigate the genetic variability of *B. tabaci* infesting sweet potato and cassava in South Sudan. Field surveys were conducted between August 2017 and July and August 2018 in 10 locations in Juba County, Central Equatoria State, South Sudan. The sequences of mitochondrial DNA cytochrome oxidase I (mtCOI) were used to determine the phylogenetic relationships between sampled *B. tabaci*. Six distinct genetic groups of *B. tabaci* were identified, including three non-cassava haplotypes (Mediterranean (MED), Indian Ocean (IO), and Uganda) and three cassava haplotypes (Sub-Saharan Africa 1 sub-group 1 (SSA1-SG1), SSA1-SG3, and SSA2). MED predominated on sweet potato and SSA2 on cassava in all of the sampled locations. The Uganda haplotype was also widespread, occurring in five of the sampled locations. This study provides important information on the diversity of *B. tabaci* species in South Sudan. A comprehensive assessment of the genetic diversity, geographical distribution, population dynamics, and host range of *B. tabaci* species in South Sudan is vital for its effective management.

## 1. Introduction

Cassava (*Manihot esculenta*) and sweet potato (*Ipomoea batatas* (L.) Lam.) are key staple root crops that assure food security in sub-Saharan Africa. This is due to their high calorie content, low production inputs, adaptation to different soil types, and resilience to climatic change as compared to other major staple food crops [1,2,3,4]. The total production of cassava in Africa amounts to 177.8 million tonnes, while that of sweet potato is 27.7 million tonnes [5]. In South Sudan, cassava is the major food security crop after maize or sorghum in the Greenbelt and the Ironstone Plateau zones, which include Western, Central, and Eastern Equatoria (Greenbelt zone), Western Bahr el Ghazal State (Ironstone Plateau zone), and Lakes State [6]. In 2015, the estimated production area for cassava was 75,910 ha and the total production was 1.1 million tonnes. However, these estimates may not represent the actual production due to the ongoing civil unrest in the country [6]. Like cassava, sweet potato is widely grown by farmers in the cassava-producing areas of South Sudan. It is the third most widely grown crop in Eastern and Western Equatoria states after cassava and groundnut [7]. Data regarding the cultivation area and production of sweet potato are not available. Agricultural surveys are mostly done by non-governmental organizations due to the ongoing war, which focus most on major staple crops, like maize, sorghum, cassava, and groundnuts.

Virus diseases is one of the major biotic factors that constrains the production of cassava and sweet potato in sub-Saharan Africa [8,9,10,11,12,13]. The most important virus diseases of cassava in sub-Saharan Africa are cassava mosaic disease (CMD) caused by cassava mosaic begomoviruses and cassava brown streak disease (CBSD) that is caused by cassava brown streak ipomoviruses [14,15,16]. The damage caused on cassava by CMD and CBSD can result in up to 82% yield losses [9,17]. CMD is prevalent wherever cassava is grown. In the 1990s, CMD was reported to be the most destructive virus disease in Western Equatoria Province in Southern Sudan before the independence of South Sudan from Sudan [18]. The cassava mosaic begomoviruses, *African cassava mosaic virus* (ACMV), *East African cassava mosaic virus* (EACMV), and *East African cassava mosaic virus*-Uganda (EACMV-UG) have been reported to be prevalent in South Sudan [7,19]. *Sweet potato chlorotic stunt virus* (SPCSV; genus *Crinivirus*) is the most important virus that affects sweet potato due to its ability to mediate severe synergistic disease with several other sweet potato-infecting viruses, which results in major yield losses [12,20,21,22]. SPCSV is a component of sweet potato virus disease (SPVD), which is the most devastating virus disease of sweet potato worldwide, being caused by co-infection of SPCSV and *Sweet potato feathery mottle virus* (SPFMV), an aphid-transmitted potyvirus [23,24,25,26]. Yield losses due to SPVD can amount to between 50% and 100% in East Africa [12,26,27]. The occurrence of SPCSV and SPFMV has been detected in South Sudan in a survey of sweet potato viruses that was conducted in 2015 in the sweet potato growing areas of Western Bahr el Ghazal, Eastern, and Central Equatoria states [28].

*Bemisia tabaci* (Gennadius) (Hemiptera: Aleyrodidae) is a polyphagous highly destructive pest of food, fiber, and ornamental crops globally [29,30]. It is known to directly damage host plants by feeding on phloem sap, and indirectly by excreting honeydew onto the surfaces of leaves, fruits, and fiber, and the transmission of plant viruses [29,31,32,33,34]. Honeydew secretion promotes the growth of sooty mold fungi (*Capnodium* spp.) on the surfaces of leaves and fruits and the contamination of cotton lint, resulting in low quality grading [32,33]. *B. tabaci* causes its most severe crop damage through vectoring plant viruses. The insect is a vector of more than 350 plant virus species belonging to five genera, including begomoviruses (family *Geminiviridae*), criniviruses (family *Closteroviridae*), ipomoviruses (family *Potyviridae*), torradoviruses (family *Secoviridae*), and some carlaviruses (family *Betaflexiviridae*). Most of the viruses transmitted are begomoviruses [30,34,35,36,37]. *B. tabaci* is a vector of cassava mosaic begomoviruses (CMBs) and cassava brown streak ipomoviruses (CBSIs) that devastate cassava crops in East and Central Africa [38,39,40]. *B. tabaci*, *B. afer* sensu lato and *Trialeurodes abutilonea* (the banded-winged whitefly) transmit SPCSV [41,42,43]. Other viruses of sweet potato that are vectored by *B. tabaci* include *Sweet potato leaf curl virus* (SPLCV), *Ipomoea leaf curl virus* (ILCV) (genus *Begomovirus*), and *Sweet potato mild mottle virus* (SPMMV) (Genus *Ipomovirus*) [43,44,45,46,47]. The outbreaks and spread of CMD and CBSD in cassava have been linked to the super-abundance of *B. tabaci* populations [9,10,48]. Rapid spread of SPCSV has also been associated with high *B. tabaci* populations, leading to high incidences of SPVD [49,50,51].

*Bemisia tabaci* is genetically complex with at least 34 cryptic species that are morphologically indistinguishable. These have been mostly identified through the sequencing of a portion of the mitochondrial cytochrome oxidase I (mtCOI) gene [52,53,54,55,56,57]. In sub-Saharan Africa, *B. tabaci* is an important vector for plant viruses that infect cassava, sweet potato, and other crops, including tomato, cucurbits, eggplant, cotton, and leguminous crops [48,58,59,60,61]. Two major groups of *B. tabaci* are known to occur in sub-Saharan Africa. One group colonizes sweet potato, vegetables, and other crops, but does not colonize cassava (non-cassava types). This group includes several putative species, including: Indian Ocean (IO), Mediterranean (MED), Middle East Asia Minor 1 (MEAM1), and Uganda [60,61]. The other cassava-colonizing group includes: Sub-Saharan Africa 1 to 5 (SSA1–5). SSA1 has been separated into five sub-groups: SSA-subgroup1 (SSA1-SG1), SSA1-SG2, SSA1-SG3, SSA1-SG4, and SSA1-SG5 [10,58,62,63,64,65].

Several molecular techniques have been used to characterize *B. tabaci* groups. They include esterase isozyme polymorphisms [66], and DNA markers, including RAPDs, PCR-RFLPs, and AFLPs [67]. Improvements in molecular tools led to the application of sequencing of 16S rRNA, cytochrome oxidase I gene portions in the mitochondrial genome, and the nuclear ribosomal intergenic spacer 1 (ITS1) [55,56,67,68]. Microsatellite markers have also been developed and used to study *B. tabaci* populations [69,70]. Recent advances have utilized SNP genotyping and nuclear genes [71,72,73]. The mitochodrial DNA cytochrome oxidase I (mtCOI) marker has been the most widely used marker for phylogenetic studies of *B. tabaci*. It has a high degree of variability and has played an important role in characterizing the genetic relationships between the *B. tabaci* cryptic species and haplotypes [53,56,74]. MtCOI has been extensively used in assessing the genetic variability, phylogeographic distribution, and identification of new invasive species of *B. tabaci* in Africa [10,60,62,75] and elsewhere [76,77,78]. However, recent SNP-genotyping using NextRAD sequencing revealed that mtCOI sequencing is not completely effective at distinguishing cassava-colonizing *B. tabaci* genotypes [71]. These were reclassified into six major groups designated: Sub-Saharan Africa East and Central Africa (SSA-ECA), Sub-Saharan Africa East and Southern Africa (SSA-ESA), Sub-Saharan Africa Central Africa (SSA-CA), Sub-Saharan Africa West Africa (SSA-WA), Sub-Saharan Africa 2 (SSA2), and Sub-Saharan Africa 4 (SSA4) [71,72]. Information on the occurrence and distribution of *B. tabaci* in sub-Saharan Africa is available, but there are no data for South Sudan. Assessing the nature of the problem that is posed by *B. tabaci* and the viruses it transmits in South Sudan and developing appropriate control strategies are currently impeded by the instability that is caused by the ongoing civil war. Data are urgently required regarding the genetic groups, haplotype diversity, geographical distribution, and the phylogenetic relationships of *B. tabaci* in South Sudan. As a first step, this study sought to address this need by sampling and characterizing *B. tabaci* collected from cassava and sweet potato in Juba County, Central Equatoria State, South Sudan. Therefore, we aimed to provide the first description of the diversity of *B. tabaci* on sweet potato and cassava in South Sudan, despite the civil unrest in the country severely limiting safe collection sites. The south of Sudan has been experiencing civil war for the past four decades, which is the reason for the lack of data in agricultural research and lack of agricultural progress. The civil unrest commenced again in 2013 one month after this project was started in the new nation of South Sudan.

## 2. Materials and Methods

### 2.1. Whitefly Sampling

Adult whiteflies (*Bemisia tabaci*) were collected from sweet potato and cassava fields across 10 locations in Juba County (Central Equatoria State, South Sudan) between July and August 2018 (Table 1 and Figure 1). The sampling was confined to these locations as a consequence of the widespread civil insecurity in many parts of South Sudan at the time of the study. Whiteflies were also sampled from tomato and squash plants adjacent to sweet potato and cassava fields. *B. tabaci* adults that were collected from sweet potato plants in greenhouses at the University of Juba in August 2017 were also added to the field collections. In total, 24 fields were sampled from the 10 locations. Whiteflies were aspirated alive and immediately preserved in 95% ethanol in vials, before being stored in the freezer at −20 °C. Sweet potato and cassava leaves that contained *B. tabaci* nymphs were cut into small pieces, put into vials, and also preserved in 95% ethanol before being stored in the freezer. *B. tabaci* were collected from several plants in each sampled field and at least 20 whiteflies were collected from each field.

### 2.2. DNA Extraction

DNA was extracted from single whiteflies, which were either adults or fourth instar nymphs. The insects were added to 3µL of lysis buffer in a 1.5 mL Eppendorf tube then macerated. The lysis buffer contained 10 mM Tris-HCl (pH 8.0, 50 mM KCL, 2.5 mM MgCl_2_, 0.45% Tween-20, 0.01% Gelatine, and 60 μg/mL Proteinase). The mixture was then vortex shaken and spun down and immediately incubated on ice for 15 min. This was followed by incubation at 55 °C in a water bath for 30 min. and the lysate was stored at −20 °C for downstream use. For PCR use, the lysate was diluted while using sterile DPEC treated water in a ratio of 1:9.

### 2.3. Mitochondrial COI (MtCOI) PCR Amplification and Sequencing

DNA extracted from 228 individual whiteflies from all the sampled locations was used for PCR amplification. Two sets of primers were used for the amplification of a partial fragment of mtCOI, primer MM1: (5′′-CTGAYATRGCKTTTCCTCG-3′-F, 5′-TTACTGCAYWTTCTGCCAC-3′-R) (IITA lab) and primer set: 2195-Bt-F (5′-TGRTTTTTTGGTCATCCRGAAGT-3′) and C012-Bt-sh2-R (5′-TTTACTGCACTTTCTGCC-3′) [79]. These primers amplified ~1300 bp and ~867 bp, respectively, portions of the mtCOI gene. The PCR reaction contained 1X QuickLoad Master Mix (New England Biolabs, UK), 1 mM MgCl_2_, 0.24 μM of each primer, 2 μL DNA, and sterile distilled water to achieve the desired reaction volume of 25 μL. PCR was carried out at 95 °C for 5 min. initial denaturation of template DNA, followed by 35 cycles at 94 °C for 40 s, 56 °C for 30 s for annealing, and 72 °C for 90 s for extension, with a final extension at 72 °C for 10 min. The PCR products were run on a 1% agorose gel in 1× TAE buffer stained with GelRed^TM^ (Biotium, Fremont, CA, USA). DNA bands were visualized while using a Gel Doc™ XR+ Gel Documentation System and only samples with intact bands were selected for sequencing. PCR products were sent to Macrogen Inc. (Rockville, MD, USA) for purification and direct sequencing. DNA sequences were manually edited while using Ridom Trace Edit v1.1.0 software (Ridom GmbH., Würzburg, Germany). The sequences were assembled into contigs using CLC Main Workbench 7.0.2 (QIAGEN, Aarhus, Denmark). Multiple alignment of edited sequences was performed while using Clustal W in MEGA version 7.0.26 [80] and the sequences were trimmed to 744 nt. Construction of a maximum-likelihood phylogenetic tree was performed using MEGA with 1000 bootstrap replicates. Sequences were blasted using GenBank’s (NCBI) Blastn and selected reference sequences with 99% to 100% identity to our mtCOI sequences were included in the phylogenetic tree for comparison with previously published haplotypes. The extent of nt sequence variation within the identified *B. tabaci* groups was examined. Estimates were obtained for the number of haplotypes, polymorphic sites (S), average number of nucleotide differences (k), nucleotide diversity (Pi), haplotype diversity (Hd), Theta per sequence and Theta per site, and significance values using the mismatch distribution procedure of Dna-SP 6.12.03 [81]. Tajima’s D and Fu’s Fs were calculated while using Dna-SP 6.12.03 to determine whether the sampled whitefly populations were stable or expanding.

## 3. Results and Discussion

*B. tabaci* whitefly samples were collected from sweet potato (*Ipomoea batatas* (L.) Lam.), cassava (*Manihot esculenta* Crantz), tomato (*Solanum lycopersicum* L.), and squash (*Cucurbita pepo* L. ‘Zucchini’) from 10 locations in Juba County, Central Equatoria State, South Sudan. The locations included Tokiman (TOK), Lologo 2 (LO2), Beitery (BET), Juba Na Bari-Jezira (JNB-JZ), Mouna Suk Hajer (MO-SH), Gudele 1 Block 5 (GU1-B5), Gudele 1 Block 7 (GU1-B7), Gudele 2 Jopa (GU2-JP), Lemon Gaba (LEG), and University of Juba (UOJ). *B. tabaci* colonizing sweet potato were collected from eight locations, except Tokiman and Gudele 1 Block 7. *B. tabaci* on cassava were sampled from five locations: Lologo 2, Juba Na Bari-Jezira, Gudele 1 Block 7, Gudele 2 Jopa, and Lemon Gaba. On tomato, *B. tabaci* were collected from three locations including Juba Na Bari-Jezira, Gudele 1 Block 5 and Gudele 1 Block 7, whereas whiteflies on squash were collected from Tokiman (Table 2). *B. tabaci* fourth instar nymphs were also collected from sweet potato in Lologo 2 and from cassava in Gudele 1 Block 7. Most of the whiteflies sampled were from sweet potato and cassava, which were the main targeted crops of this study. Consequently, the whiteflies that were collected from tomato and squash were from fields adjacent to either sweet potato or cassava plantings. The number and distribution of collection sites varied, depending on the number of fields that were found in each location and the relative abundance of *B. tabaci* in those fields. However, most importantly, the sampling provided a representative collection of *B. tabaci* from the major target crops (cassava and sweet potato). The degree of variability subsequently revealed by the sequencing and phylogenetic analysis confirmed this point.

In total, 183 whitefly samples were sequenced, out of which 162 produced high quality mtCOI sequences. There was a high level of diversity among *B. tabaci* populations that were collected from the sampled crop plants. The sequences obtained from sweet potato, tomato, and squash were grouped into three phylogenetically distinct groups, which included (MED), Indian Ocean (IO) and Uganda. The sequences from cassava were grouped into three distinct groups, SSA1-SG1, SSA1-SG3, and SSA2 (Figure 2). These groups were identified based on the topology of the phylogenetic tree and the clustering of the sequences that were obtained from this study relative to the reference sequences retrieved from GenBank. The predominant haplotype MED had a total of 90 whiteflies, which accounted for 55.5% of all the whiteflies collected from the four host plants. Of these, 72 whiteflies (44.4%) were found on sweet potato (Table 3). The second most abundant haplotype was SSA2 with 43 whiteflies (26.5%), all of them being found on cassava. SSA1-SG1 was a second haplotype found only on cassava for which there were 13 whiteflies (8%). The other haplotypes were Uganda, which was present on sweet potato and had a total of 11 whiteflies (6.8%), Indian Ocean with four whiteflies (2.5%) and SSA1-SG3, which was the least frequent haplotype with only one whitefly (0.6%) found on sweet potato (Table 3). A total of 45 selected sequences that represent haplogroups found in this study have been submitted to GenBank under the following accession names (MN318379-MN318423).

The clustering of the whiteflies SSA2, SSA1-SG1, and SSA1-SG3 into a distinct major clade separate from *B. tabaci* whiteflies that do not colonize cassava is consistent with what has been reported in other studies of *B. tabaci* from various cassava-growing countries in Africa [60,61,71]. The grouping of MED and Indian Ocean haplotypes is also consistent with what has been reported in previous studies [60,61]. Uganda, which was depicted by a clearly defined monophyletic grouping in our mtCOI sequence analysis, has previously been identified as a genetically distinct haplotype that occurs in East Africa [58,70].

We found that *B. tabaci* MED was predominant on sweet potato, tomato, and squash in all of the sampled locations. MED is a globally important *B. tabaci* haplotype group which is thought to have originated from Africa. Consequently, there are numerous other reports of its prevalence on a wide range of crop and weed hosts [59,60,64]. *B. tabaci* MED has been reported to be extremely polyphagous and invasive [54], causing damage to both field and greenhouse crops [82]. It has also developed resistance to various insecticides under intensive production systems [83,84,85]. The presence of *B. tabaci* MED in all locations and on all sampled crop plants in our study in South Sudan suggests that this haplotype is an important pest of sweet potato and other crops in Juba County. MED has been reported to widely occur in sub-Saharan Africa, and it seems likely, therefore, that, in addition to Juba County, it is an important *B. tabaci* haplotype throughout South Sudan. Moreover, as SPCSV transmitted by *B. tabaci* is one of the most important viruses affecting sweet potato in this region of East Africa [12], it is likely that this is the main vector of this virus in South Sudan. However, future investigations should determine the relative abilities to transmit SPCSV of each of the three *B. tabaci* haplotypes occurring on sweet potato. Since no similar studies have been conducted anywhere else in sub-Saharan Africa, this represents an important gap in the existing understanding of the relationship between *B. tabaci* haplotype groups and the viruses that they vector.

The MED haplotype analyses revealed six haplotypes amongst the samples that were collected from South Sudan (Table 4). Two of these are previously described African MED haplotypes, whilst the other four are new unique haplotypes that fall within the MED group. Haplotype diversity (0.51), nucleotide diversity (0.012), and a positive significant Tajima’s D (2.07283: *p* < 0.05) suggest that the population is undergoing balancing selection and has not undergone rapid recent expansion. Sixty-two of the 90 MED sequences (Haplotype 1) represent an important African MED haplotype, for which there are a further 17 sequences in GenBank from Cameroon, Uganda, and Nigeria. The samples in this haplotype were predominantly from sweet potato, although there were also individuals from tomato, squash, and cassava, which indicated that it could be sharing host plants. Haplotype 2 had eight sequences from sweet potato that were identical to 13 sequences from GenBank originating from Sudan, Cameroon, Uganda and Burkina Faso. Haplotype 3 had eight sequences from sweet potato and squash. These were most closely matched (99.7%) with a GenBank sequence from Uganda KX570768. Haplotypes 4 and 5 occurred on sweet potato and had eight and three sequences, respectively. These were most closely related (99.7%) to a sequence from China (MH908653). Haplotype 6, which was recorded from sweet potato, had one sequence sharing 99.9% homology with MH908653 from China. Currently GenBank hosts 944 MED sequences that comprise 168 haplotypes. 673 (71%) of the sequences cluster in three major haplotypes that are spread worldwide. Of the 168 haplotypes, 137 (81%) have only one sequence in GenBank, although it is possible that some of these may be erroneously considered as unique haplotypes due to the frequent occurrence of sequencing errors in mtCOI data submitted to this database. In some scenarios, mitochondrial bar-coding has been found to overestimate the number of species or scale of divergence, where there are nuclear mitochondrial DNA pseudogenes (NUMTs) that can be PCR-amplified with mitochondrial primers [86]. A study has demonstrated that NUMTs were the cause for the incorrect identification of a putative *Bemisia* species that was given the name MEAM2 based on mtDNA COI data [87]. Another study using whole genome nuclear markers on the major clades of *B. tabaci* revealed the existence of fewer putative species (five so far), as opposed to the much larger number reported with mtCO1 [73].

In this study, *B. tabaci* Indian Ocean were collected from sweet potato and tomato, and their sequences were most closely related to Reunion 1 from Spain [88], although *B. tabaci* Indian Ocean has also been widely reported from sub-Saharan Africa and the surrounding islands [60,65,89]. Haplotype analysis revealed the existence of two Indian Ocean haplotypes. The Uganda haplotype sequences were obtained from several whiteflies that were collected from sweet potato and one individual from tomato. The South Sudan ‘Uganda’ haplotype sequences were identical to the original Uganda haplotype sequence also obtained from a whitefly adult collected from sweet potato in Uganda (33NamSP-AY057174) [58]. However, Sseruwagi et al. [60] reported the occurrence of sweet potato Uganda haplotype on crop plants other than sweet potato, and Wainana [90] made similar observations from western Kenya, noting the presence of the Uganda haplotype on common bean as well as sweet potato. These data suggest that this haplotype is confined to East Africa and it has a relatively narrow host range, specializing on sweet potato. Our results represent the northernmost record of haplotype ‘Uganda’ and the third country report.

In the studied cassava group of *B. tabaci*, the largest number of samples were SSA2, and these were distributed through all of the locations. Two SSA2 haplotypes were identified (Table 4). Haplotype diversity (0.509), nucleotide diversity (0.00205), and a positive significant Tajima’s D (257824: *p* < 0.05) suggest that the population has not undergone recent expansion, but is instead experiencing balancing selection. SSA1-SG1 was less frequent, as it was only detected at two locations and comprised only one haplotype. These results differ from other recent findings from East and Central Africa, which have shown SSA1-SG1 to be the predominant *B. tabaci* haplotype on cassava [10,61,71]. SSA2, which was previously associated with the severe CMD epidemic in Uganda [58], has been reported to be absent in more recent whitefly collections from cassava in Uganda and western Kenya [65,91], and replaced by SSA1-SG1 [10], although low frequencies of this haplotype were reported from Uganda and Kenya between 2004 and 2010. Recent studies have noted the occurrence of SSA2 on cassava in western Kenya and weedy hosts in Uganda [72,79,92]. The detection of SSA1-SG1 and SSA2 fourth instar nymphs of *B. tabaci* confirms that both of the haplotypes colonize cassava in South Sudan. A recent continent-wide assessment of cassava-colonizing *B. tabaci* in sub-Saharan Africa noted that SSA2 was the most widely distributed of the recorded haplotypes [72]. Significantly, this haplotype co-occurs with others throughout its geographic range (stretching from Sierra Leone in West Africa to Kenya in the East), but it appears to be less frequent than SSA1 haplotypes in all cases. Our data suggest that South Sudan could be an exception to this pattern, since there were more than three times as many SSA2 individuals recorded when compared to those of SSA1-SG1.

In the 1990s, CMD was reported to be highly destructive in the Western Equatoria Province of pre-independence southern Sudan [18]. Furthermore, in a baseline survey that was conducted on cassava in 2005 in Eastern and Western Equatoria states, African cassava mosaic virus (ACMV), East African cassava mosaic virus (EACMV), and East African cassava mosaic virus-Uganda (EACMV-UG) were found to be the viruses that affect cassava [7]. SSA2 was shown to be the most abundant *B. tabaci* haplotype in areas that were affected by the severe CMD epidemic, which spread through Uganda in the 1990s. It is quite likely that there might have been a similar association between virus and vector in southern Sudan during this period. However, whilst SSA1-SG1 subsequently displaced SSA2 as the predominant *B. tabaci* haplotype on cassava in Uganda, this change might not have happened further north in southern Sudan, with the result being that SSA2 is currently the main cassava-colonizing *B. tabaci* haplotype in present day South Sudan. The reasons behind these contrasting patterns of population change in Uganda and South Sudan are not currently apparent, but would be a useful topic for future study.

In this study, a single individual of non-cassava *B. tabaci* haplotype MED was collected from cassava. These rare occurrences have been reported elsewhere [61,75,91]. However, previous studies have demonstrated that non-cassava *B. tabaci* whiteflies are unable to reproduce on and colonize cassava [93], partly since they are unable to feed effectively on cassava plant hosts [94]. In each of these instances, it has been concluded that whiteflies of non-cassava *B. tabaci* haplotypes occurring on cassava are present as visitors, and are not colonizing the crop.

## 4. Conclusions

This study presents the first report on the genetic diversity of *B. tabaci* whitefly populations collected from South Sudan. Six *B. tabaci* haplotype groups, which include three non-cassava groups (MED, Indian Ocean and Uganda) and three cassava groups (SSA1-SG1, SSA1-SG3 and SSA2), were identified. MED and SSA2 were the most prevalent and most widely distributed amongst the sampled locations. The Uganda haplotype is also widespread and it was identified from five of the locations. The discovery of six *B. tabaci* haplotype groups from the relatively small portion of South Sudan that was sampled does suggest that, like Uganda, this part of East Africa has a high level of whitefly diversity. This provides a strong indication that this part of Africa might have been a source for MED whiteflies that have had devastating global impacts as an invasive pest [95]. It is also significant that the MED species group of *B. tabaci* includes some of the most insecticide-resistant populations of whiteflies. Therefore, any future management efforts will need to apply extreme caution in the application of chemical insecticides to preclude the development of whitefly resistance, although *Bemisia* whiteflies may not be present on sweet potato and other host plants at high abundance levels. Whitefly populations that were observed in South Sudan were associated with transmission of viruses causing damaging disease in cassava and sweet potato. Improving the understanding of the dynamic interactions between vector and virus will be important for each of these crop-virus-vector pathosystems. An essential first step in this task will be conducting a comprehensive assessment of the genetic diversity, geographical distribution, population dynamics, and host range of *B. tabaci* species in South Sudan. This new knowledge will then provide the basis for the development of effective whitefly management strategies.

## Figures and Tables

**Figure 1 insects-11-00058-f001:**
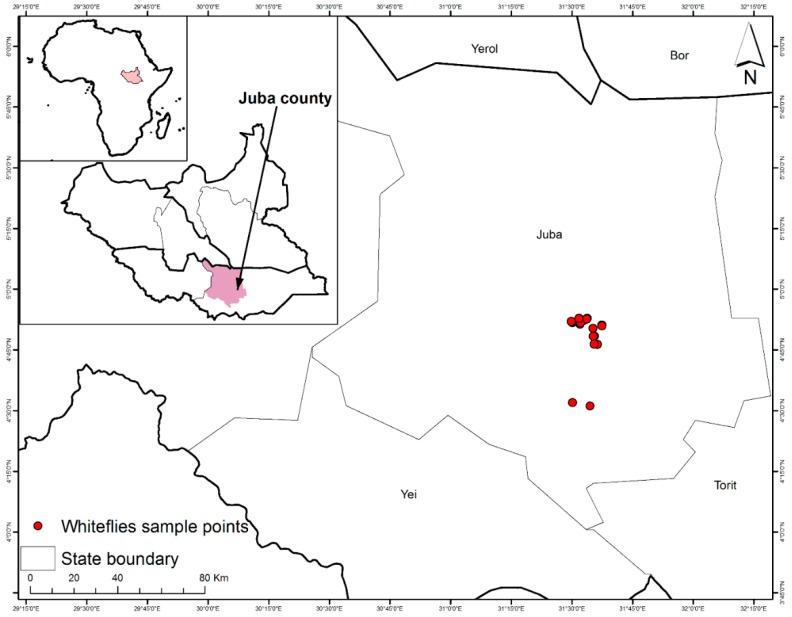
Sampling sites of *Bemisia tabaci* haplotypes on sweet potato, cassava, tomato and squash in Juba County, Central Equatoria State, South Sudan.

**Figure 2 insects-11-00058-f002:**
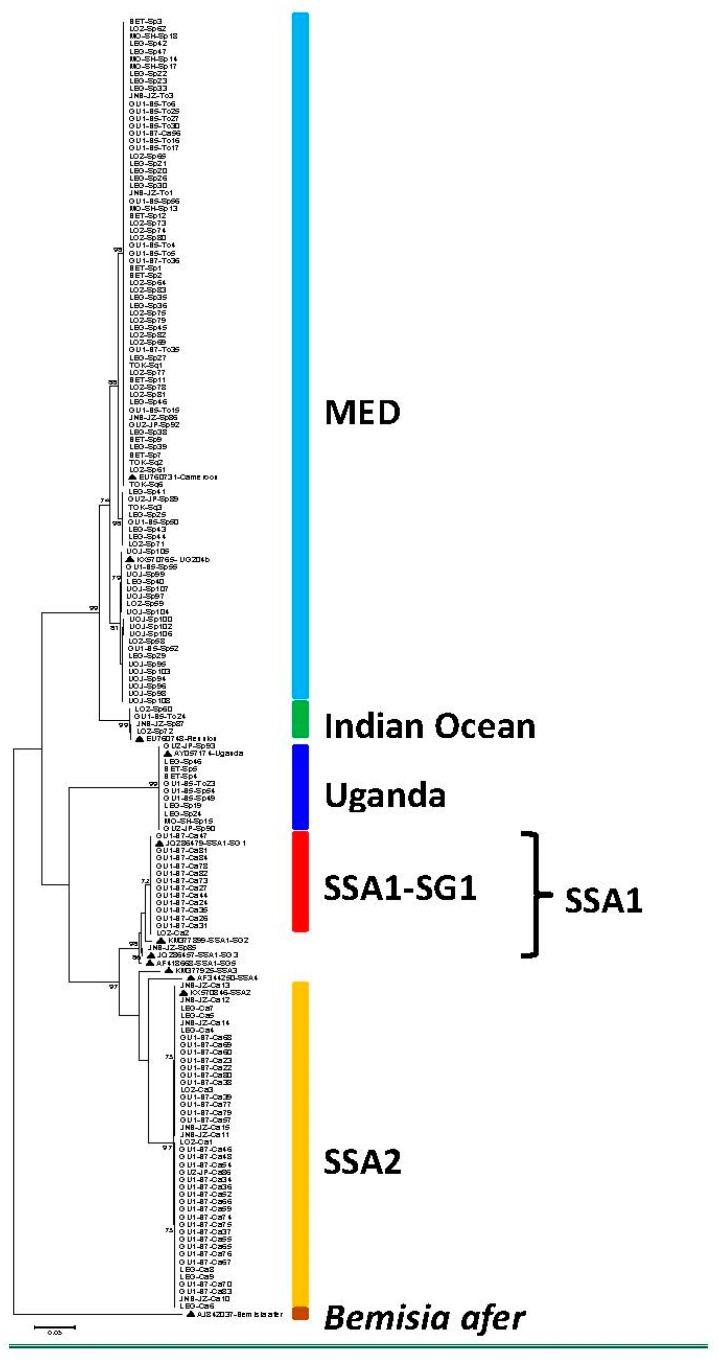
Maximum likelihood phylogenetic tree constructed for mtCOI sequences obtained from *Bemisia tabaci* collected from 10 locations in Juba County, central Equatoria State, South Sudan between July and August 2018. Reference sequences from GenBank (▲) are included for comparison.

**Table 1 insects-11-00058-t001:** Geographical information of sampling sites in Juba County, Central Equatoria State, South Sudan.

Host Plant	Area/Payam	Sampling Site	Latitude (°N)	Longitude (°E)	Elevation (masl)	Date
Cassava	Rajaf West	Lologo 2	04°48.456′	031°35.463′	468	31.07.2018
	Northern Bari	Lemon Gaba	04°52.051′	031°29.873′	487	03.08.2018
	Kondokoro	Juba Na Bari-Jezira	04°51.167′	031°37.430′	457	04.08.2018
	Munuki	Gudele 1 Block 7	04°52.383′	031°33.025′	478	04.08.2018
	Northern Bari	Gudele 2 Jopa	04°52.852′	031°31.742′	472	04.08.2018
Tomato	Kondokoro	Juba Na Bari-Jezira	04°50.945′	031°37.392′	460	04.08.2018
	Munuki	Gudele 1 Block 5	04°52.849′	031°33.770′	466	27.07.2018
	Munuki	Gudele 1 Block 7	04°52.383′	031°33.025′	478	04.08.2018
Squash	Rajaf East	Tokiman	04°46.396′	031°36.334′	460	04.08.2018
Sweet potato	Rajaf West	Beitery	04°51.323′	031°32.009′	508	19.07.2018
	Munuki	Mouna-Suk Hajer	04°31.130′	031°34.483′	490	24.07.2018
	Northern Bari	Lemon Gaba	04°51.688′	031°30.104′	506	25.07.0218
	Northern Bari	Lemon Gaba	04°31.965′	031°30.140′	490	27.07.2018
	Munuki	Gudele 1 Block 5	04°52.849′	031°33.770′	466	27.07.2018
	Northern Bari	Lemon Gaba	04°51.850′	031°30.083′	498	28.07.2018
	Rajaf West	Beitery	04°51.669′	031°32.117′	488	28.07.2018
	Rajaf West	Lologo 2	04°48.443′	031°35.564′	473	30.08.3018
	Rajaf West	Lologo 2	04°46.447′	031°35.496′	476	31.07.2018
	Rajaf West	Lologo 2	04°48.344′	031°35.302′	474	01.08.2018
	Munuki	Gudele 1 Block 5	04°52.662′	031°33.690′	469	02.08.2018
	Northern Bari	Lemon Gaba	04°52.051′	031°29.873′	487	03.08.2018
	Kondokoro	Juba Na Bari-Jezira	04°50.945′	031°37.392′	460	04.08.2018
	Northern Bari	Gudele 2 Jopa	04°52.766′	031°31.757′	470	04.08.2018
	Juba	University of Juba (Green house)	04°50.327′	031°35.225′	494	10.08.2017

**Table 2 insects-11-00058-t002:** Number of *B. tabaci* sequences obtained from sampled locations and host plants in Juba County, Central Equatoria State, South Sudan.

Location	Host Plant				
	Sweet Potato (Sp)	Cassava (Ca)	Tomato (To)	Squash (Sq)	Total
Tokiman (TOK)	-	-	-	4	4
Lologo 2 (LO2)	20	3	-	-	23
Beitery (BET)	9	-	-	-	9
Juba Na Bari-Jezira (JNB-JZ)	3	6	2	-	11
Mouna-Suk Hager (MO-SH)	5	-	-	-	5
Gudele 1 Block 5 (GU1-B5)	6	-	11	-	17
Gudele 1 Block 7 (GU1-B7)	-	41	2	-	43
Gudele 2 Jopa (GU2-JP)	4	1	-	-	5
Lemon Gaba (LEG)	25	6	-	-	31
Univeristy of Juba (UOJ)	14	-	-	-	14
Total	86	57	15	4	162

**Table 3 insects-11-00058-t003:** *Bemisia tabaci* haplotype groups (numbers and percentages) on four host plants from Juba County, Central Equatoria State, South Sudan.

*B. tabaci* Haplotypes	Host Plants				
	Sweet Potato	Cassava	Tomato	Squash	Total
MED	72 (44.4%)	1 (0.6%)	13 (8.0%)	4 (2.5%)	90 (55.5%)
Indian Ocean	3 (1.9%)	-	1 (0.6%)	-	4 (2.5%)
Uganda	10 (6.2%)	-	1 (0.6%)	-	11 (6.8%)
SSA1-SG1	-	13 (8.0%)	-	-	13 (8.0%)
SSA1-SG3	1(0.6%)				1(0.6%)
SSA2	-	43 (26.5%)	-	-	43 (26.5%)
Total	86	57	15	4	162

**Table 4 insects-11-00058-t004:** Population genetic analysis of *Bemisia tabaci* groups from Juba county, Central Equatoria region, South Sudan.

Parameter	All	MED	SSA2	SSA1-SG1	Uganda	Indian Ocean	SSA1-SG3
Sample size	162	90	43	13	11	4	1
Number of haplotypes	13	6	2	1	1	2	1
Polymorphic sites (S)	212	28	3	0	0	1	-
Average number of nucleotide differences (k)	73.45702	9.26367	1.52824	0.000	0.000	0.66667	-
Nucleotide diversity (Pi)	0.09873	0.01245	0.00205	0.000	0.000	0.00090	-
Haplotype diversity (Hd)	0.804	0.506	0.509	0.000	0.000	0.667	-
Variance of Hd	0.00054	0.00356	0.00039	0.000	0.000	0.04167	-
Standard deviation of Hd	0.023	0.060	0.020	0.000	0.000	0.204	-
Theta per sequence	45.21592	5.52109	0.69336	-	-	0.54545	-
Theta per site	0.06077	0.00742	0.00093	0.000	0.000	0.00073	-
Fu’s Fs statistic	96.201	16.737	5.448	-	-	0.540	-
Tajima’s D *p*	2.01806 0.10 > *p* > 0.05	2.07283 *p* < 0.05	2.57824 *p* < 0.05	-	-	1.63299 *p* > 0.10	-

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
