# Peer review of "Genetic Diversity of Bemisia tabaci (Gennadius) (Hemiptera: Aleyrodidae) Colonizing Sweet Potato and Cassava in South Sudan"

_insects, 2020, doi:10.3390/insects11010058_

Round 1

Reviewer 1 Report

The authors have presented a good study of the genetic diversity in Bemisia tabaci (Gennadius), a pest causing devastations to production of cassava and sweet potato in Sudan by surveying 10 locations in Juba county, south Sudan. The manuscript is very well written. The flow of information is also very clear. Here are some suggestions on improving this article;

Introduction

Lines 30-44 –An additional map showing the growing regions of cassava and if possible, of sweet potato in Sudan would be great to have.

Lines 45-80- On the map requested above or in a separate map, could you also add the range for B. tabaci, along with incidence reported for the devastating diseases caused by the viruses that B. tabaci acts as a vector for.

Lines 92-93- Are there other markers such as microsatellites that are used to determine genetic variability? If so, do the species of B. tabaci hold up, when analyses are done using markers other than mtCOI? Please provide a sentence or two covering these questions. The information on SNP markers reveals that mtCOI sequencing isn’t adequate, however, there may be microsatellite markers that could provide better resolution or you can sequence the SNP markers region on these samples to get the same information.

The Material and Methods

Line 114-115- Why were only the 10 locations in Juba county selected? Is it because this is the only region affected by whiteflies, or are the only location where the crops are grown? The map requested in Introduction, should help clarify some of these points. Please provide a rationale for your sampling strategy.

Results and Discussion

In Table 2 Can you indicate at which of the locations the sequencing of the whiteflies yielded poor results? I ask because some of the locations do not have any whiteflies samples on cassava and sweet potato.

Also, the total number of samples collected per location vary widely. Is this due to the limitation in number of samples that were available due to differing incidence of B. tabaci in these locations? Please explain.

Lines 217-218- “..this haplotype is an important pest of sweet potato and other crops in Juba County and likely also in other parts of South Sudan.” Without sampling and actual analysis the underlined statement shouldn’t be in there.

Lines 241-243-“Of the 168 haplotypes, 137 (81%) have only one sequence in GenBank, although it is possible that some of these may erroneously be considered as unique haplotypes due to the frequent occurrence of sequencing errors in mtCOI data submitted to this database”

Do you have a reference to validate the statement about frequent occurrence of sequencing errors in mtCOI data in GenBank? If so, please add that here.

Author Response

Review 1 Report

The authors have presented a good study of the genetic diversity in Bemisia tabaci (Gennadius), a pest causing devastations to production of cassava and sweet potato in Sudan by surveying 10 locations in Juba county, South Sudan. The manuscript is very well written. The flow of information is also very clear. Here are some suggestions on improving this article;

Introduction

Lines 30-44 –An additional map showing the growing regions of cassava and if possible, of sweet potato in Sudan would be great to have.

Thanks very much for this comment. It is true that a map showing the distributions of cassava and sweet potato in South Sudan would be interesting, however, we considered this option but thought that it might be simplest just to explain in the text within a few short lines to save space. We trust that this is acceptable. In the text, you will note the descriptions of the distributions of the two crops in lines 34-41.

Lines 45-80- On the map requested above or in a separate map, could you also add the range for B. tabaci, along with incidence reported for the devastating diseases caused by the viruses that B. tabaci acts as a vector for.

This is certainly a valuable suggestion, but these aspects were unfortunately outside of the scope of the current study. As described in the text, B. tabaci is a pantropical insect pest that occurs throughout tropical and sub-tropical parts of sub-Saharan Africa. This means that it will also occur throughout South Sudan. However, there have not yet been any studies to describe the distribution of B. tabaci and its different genotypes in South Sudan. Our objective for this paper was to provide the first molecular characterisation of whiteflies in South Sudan, and we are confident that we achieved this objective. Future work will aim to examine the distributions of B. tabaci genotypes as well as the viruses that they transmit.

Lines 92-93- Are there other markers such as microsatellites that are used to determine genetic variability? If so, do the species of B. tabaci hold up, when analyses are done using markers other than mtCOI? Please provide a sentence or two covering these questions. The information on SNP markers reveals that mtCOI sequencing isn’t adequate, however, there may be microsatellite markers that could provide better resolution or you can sequence the SNP markers region on these samples to get the same information.

Thank you for the suggestion. We have added a paragraph in the manuscript on other molecular techniques for characterization of Bemisia tabaci. The SNP markers and nuclear genes from recent studies show mtCOI is not completely effective but these recent tools are costly for routine characterization of Bemisia tabaci. In spite of the imperfection of the mtCOI diagnostic, it continues to be the most affordable and widely used whitefly molecular characterisation tool. We therefore think that the data provided in this paper represent an important first step in characterizing whiteflies in South Sudan.

The Material and Methods

Line 114-115- Why were only the 10 locations in Juba county selected? Is it because this is the only region affected by whiteflies, or are the only location where the crops are grown? The map requested in Introduction, should help clarify some of these points. Please provide a rationale for your sampling strategy.

Thank you. Due to inevitable limitations imposed by civil war in South Sudan, this research could not cover all cassava and sweet potato producing areas of South Sudan. South Sudan has been in a state of war for decades, that led to war-related destruction and insecurity, and population displacements countrywide. This has hindered access to technological advances in crop production, and research of crop plants in South Sudan. As such, not much has been done to identify and manage pest and diseases affecting crop plants as well as insect vectors of plant viruses in South Sudan of which cassava and sweet potato are some of the most important. There is currently no information available on the occurrence and distribution of B. tabaci in South Sudan. There is a need to assess the problem posed by B. tabaci and the viruses it transmits in South Sudan and developing appropriate control strategies. Data is urgently required on the genetic groups, haplotype diversity, geographical distribution and the phylogenetic relationships of B. tabaci in South Sudan.  Unfortunately, the civil insecurity and instability caused by the ongoing civil war in the country has made it impossible to collect data in the crop producing areas of South Sudan. At the time of the study reported here, data were accessible for only one geographical location, Juba County in Central Equatoria State where there was relative security. Even this work represents a major (and risky) survey achievement. As a first step, we aimed to provide the first description of the diversity of B. tabaci on sweet potato and cassava in South Sudan in the accessible locations. The ten locations sampled were representative of the accessible area. One of the fascinating things about the results obtained were the high level of diversity within this sampled area. This is the reason behind our statement about the high degree of whitefly diversity from this part of Africa (line 338).

Results and Discussion

In Table 2 Can you indicate at which of the locations the sequencing of the whiteflies yielded poor results? I ask because some of the locations do not have any whiteflies samples on cassava and sweet potato.

Most of the samples obtained provided high quality sequencing results (162 out of 183). It is very normal in mtCOI sequencing to have a proportion of samples that do not yield high quality sequences. This can be readily verified by checking other publications on mtCOI characterisation of whiteflies. Not all locations yielded whitefly samples for all crops. This primarily reflects the presence of different crop plants at different locations. The sampling of different crops at different locations is summarized in Table 1.

Also, the total number of samples collected per location vary widely. Is this due to the limitation in number of samples that were available due to differing incidence of B. tabaci in these locations? Please explain.

Thank you for the comment. Yes, the number and distribution of collection sites varied depending on the number of fields that were found in each location and the incidence of B. tabaci in those fields. However, the major limitation that led to this variability was insecurity. In spite of these limitations, however, we consider that the mtCOI sequence variability obtained is likely to be a strong representation of what is present in the sampled zone. Both the results for cassava and non-cassava colonizing B. tabaci revealed the occurrence of the haplotype groups that might have been anticipated. In fact, the variability identified, particularly within haplotype groups, was even more than expected.  

Lines 217-218- “..this haplotype is an important pest of sweet potato and other crops in Juba County and likely also in other parts of South Sudan.” Without sampling and actual analysis the underlined statement shouldn’t be in there.

This section in the text has been modified to address this point.

Lines 241-243-“Of the 168 haplotypes, 137 (81%) have only one sequence in GenBank, although it is possible that some of these may erroneously be considered as unique haplotypes due to the frequent occurrence of sequencing errors in mtCOI data submitted to this database”

Do you have a reference to validate the statement about frequent occurrence of sequencing errors in mtCOI data in GenBank? If so, please add that here.

Thank you for the suggestion. We have added a paragraph in the manuscript with references to explain this point more clearly.

Reviewer 2 Report

The report is a useful contribution to our knowledge of this pest insect which in fact is a collection of several cryptic species with different host plant association and different roles as disease vector.

It is well written.

Author Response

Thank you very much for this encouraging comment. We have no further response to add from the authors' side.

Reviewer 3 Report

I would be hesitant to accept this paper in a current form. I have a few major concerns.

I am inclined to consider phylogenetic tree redundant. The idea of the study is not to explore the phylogenetic or evolutionary relationships among these cryptic species. Bemisia literature has leaning towards a premature but close to an agreement in accepting these haplotypes groups as cryptic species. Authors can easily use consensus sequences of already established cryptic species to put names on their sequences (as suggested in Ahmed et al 2012, Journal of applied entomology). However, any sequence that does not fall in between the boundaries of Bemisia consensus sequences of the respective cryptic species should be discussed and compared with sequences published in other studies.

There is a sequence in GenBank from Sudan, (Accession Number#EU760727), which was placed in New World cryptic species, though it was unconnected in genetic network of New Word cryptic species in De Barro and Ahmed 2011, Plos One. Authors may also want to bring this in the discussion and analysis.

There were 23 unique MED haplotypes groups until 2011 according to De Barro and Ahmed 2011 (Plos One). One haplotypes group (GenBank Accession Number#DQ133378) that comes from Burkina Faso, Cameroon, Sudan) and four singleton haplotype groups (GenBank Accession Numbers AY827612-15) were found from Sudan. I would also analyze and discuss which of these match with the MED haplotypes authors found in their current study.

De Barro and Ahmed 2011 (Plos One) pointed out the possibility that MED has moved back to Sub-Saharan Africa from the Mediterranean region, possibly via Sudan and that this shift is coincident with the global spread of invasive MED that has been documented over the past 10 years. I would not ignore this possibility and try to reject or accept it by comparing all MED sequences from this study with 23 MED unique haplotypes groups representative in the study by De Barro and Ahmed 2011.

Author Response

Review 3 Report

I would be hesitant to accept this paper in a current form. I have a few major concerns.

I am inclined to consider phylogenetic tree redundant. The idea of the study is not to explore the phylogenetic or evolutionary relationships among these cryptic species. Bemisia literature has leaning towards a premature but close to an agreement in accepting these haplotypes groups as cryptic species. Authors can easily use consensus sequences of already established cryptic species to put names on their sequences (as suggested in Ahmed et al 2012, Journal of applied entomology). However, any sequence that does not fall in between the boundaries of Bemisia consensus sequences of the respective cryptic species should be discussed and compared with sequences published in other studies.

Thank you for your suggestion. We would, however, like to keep the phylogenetic tree since it is the first to be published for B. tabaci diversity in South Sudan. Phylogenetic trees are still being widely used, for example Tocko-et al. 2017 “Genetic diversity of Bemisia tabaci species colonizing cassava in Central African Republic characterized by analysis of cytochrome c oxidase subunit I. PLoS ONE. 12 (8), e0182749; Ally et al. 2019 "What has changed in the outbreaking populations of the severe crop pest whitefly species in cassava in two decades?." Scientific reports 9: 1-13.; Mugerwa et al. 2018 "African ancestry of New World, Bemisia tabaci-whitefly species." Scientific reports 8: 2734.

All of our sequences fell within the boundaries of previously described cryptic species. In the phylogenetic tree, we therefore used reference sequences for each of the main cryptic species groups for comparison. We found the variability amongst sequences in each major cryptic species group fascinating, which is why we examined this with further statistics in the paper. The tree is a nice way for readers of the paper to visualize the high degree of variability that we discovered through this study, particularly within the MED group.

There is a sequence in GenBank from Sudan, (Accession Number#EU760727), which was placed in New World cryptic species, though it was unconnected in genetic network of New Word cryptic species in De Barro and Ahmed 2011, Plos One. Authors may also want to bring this in the discussion and analysis.

Thank you for suggestion. This sequence was included among samples collected from South Sudan. It does not cluster with any of the samples in this study. We therefore did not see the need to discuss this sequence in this study.

There were 23 unique MED haplotypes groups until 2011 according to De Barro and Ahmed 2011 (Plos One). One haplotypes group (GenBank Accession Number#DQ133378) that comes from Burkina Faso, Cameroon, Sudan) and four singleton haplotype groups (GenBank Accession Numbers AY827612-15) were found from Sudan. I would also analyze and discuss which of these match with the MED haplotypes authors found in their current study.

Thank you for suggestion. Eight samples from our study are the DQ133378 group which is already represented in the phylogenetic tree by reference sequence KX570765-UG204b from Uganda. None of the samples in this study clustered with any of the GenBank accessions AY827612-15. These four sequences still remained singletons even with the current total of 1034 MED sequences in GenBank.

De Barro and Ahmed 2011 (Plos One) pointed out the possibility that MED has moved back to Sub-Saharan Africa from the Mediterranean region, possibly via Sudan and that this shift is coincident with the global spread of invasive MED that has been documented over the past 10 years. I would not ignore this possibility and try to reject or accept it by comparing all MED sequences from this study with 23 MED unique haplotypes groups representative in the study by De Barro and Ahmed 2011.

Thank you for suggestion. Haplotypes were computed from a total of 1034 MED sequences using DnaSP6 software. The sequences comprised a combination of those deposited in GenBank and those from this study. These sequences (618 bp) were grouped into 171 haplotypes, out of which 137 were represented by single sequences. The South Sudan samples clustered into five haplotypes out of the 171. Two of the haplotypes had matching sequences in the GenBank represented by EU760731 and KX570765-UG204b. Three haplotypes were unique, and they are represented in the phylogenetic tree with first sequences LEG-Sp41, UOJ-Sp100 and LO2-Sp58 respectively. None of the MED haplotypes that are the most frequently represented in GenBank, and which have been associated with invasive spread, were identified from this study. We tried to emphasize the point in the manuscript that the MED haplotypes identified in South Sudan were either the same as those reported from other parts of African (therefore likely African in origin) or were unique. We also noted that the occurrence of several unique MED haplotypes from our study in South Sudan emphasizes the likelihood that sub-Saharan Africa is a centre of diversity for the MED haplogroup.

Round 2

Reviewer 1 Report

I am happy with the justification provided by authors on most comments. Here is my recommendation based on information provided by the authors.

Material and Methods

Line 114-115- Why were only the 10 locations in Juba county selected? Is it because this is the only region affected by whiteflies, or are the only location where the crops are grown? The map requested in Introduction, should help clarify some of these points. Please provide a rationale for your sampling strategy.

Thank you. Due to inevitable limitations imposed by civil war in South Sudan, this research could not cover all cassava and sweet potato producing areas of South Sudan. South Sudan has been in a state of war for decades, that led to war-related destruction and insecurity, and population displacements countrywide. This has hindered access to technological advances in crop production, and research of crop plants in South Sudan. As such, not much has been done to identify and manage pest and diseases affecting crop plants as well as insect vectors of plant viruses in South Sudan of which cassava and sweet potato are some of the most important. There is currently no information available on the occurrence and distribution of B. tabaci in South Sudan. There is a need to assess the problem posed by B. tabaci and the viruses it transmits in South Sudan and developing appropriate control strategies. Data is urgently required on the genetic groups, haplotype diversity, geographical distribution and the phylogenetic relationships of B. tabaci in South Sudan.  Unfortunately, the civil insecurity and instability caused by the ongoing civil war in the country has made it impossible to collect data in the crop producing areas of South Sudan. At the time of the study reported here, data were accessible for only one geographical location, Juba County in Central Equatoria State where there was relative security. Even this work represents a major (and risky) survey achievement. As a first step, we aimed to provide the first description of the diversity of B. tabaci on sweet potato and cassava in South Sudan in the accessible locations. The ten locations sampled were representative of the accessible area. One of the fascinating things about the results obtained were the high level of diversity within this sampled area. This is the reason behind our statement about the high degree of whitefly diversity from this part of Africa (line 338).

Please add a sentence or two with information on civil insecurity and instability which limited your ability to collect data outside of Juba county in addition to Lines 114-115. I ask this of you as many won't be familiar with the extent or geographic regions affected by the civil war. 

 Results and Discussion

Also, the total number of samples collected per location vary widely. Is this due to the limitation in number of samples that were available due to differing incidence of B. tabaci in these locations? Please explain.

Thank you for the comment. Yes, the number and distribution of collection sites varied depending on the number of fields that were found in each location and the incidence of B. tabaci in those fields. However, the major limitation that led to this variability was insecurity. In spite of these limitations, however, we consider that the mtCOI sequence variability obtained is likely to be a strong representation of what is present in the sampled zone. Both the results for cassava and non-cassava colonizing B. tabaci revealed the occurrence of the haplotype groups that might have been anticipated. In fact, the variability identified, particularly within haplotype groups, was even more than expected. 

Please add your explanation on variability here in form of a note to the table or any other suitable format.
